# User Experience of Older People While Using Digital Health Technologies: A Systematic Review

**Eiko Takano** [1,*], **Hisataka Maruyama** [2], **Tetsuta Takahashi** [3], **Kouki Mori** [2], **Kota Nishiyori** [3], **Yoshifumi Morita** [3], **Toshio Fukuda** [4], **Izumi Kondo** [1] and **Yutaka Ishibashi** [3]

1    Assistive Robotics Center, National Center for Geriatrics and Gerontology, Obu 474-8511, Japan; ik7710@ncgg.go.jp
2    Department of Micro Nano Mechanical Science and Engineering, Nagoya University, Nagoya 464-8603, Japan; maruyama.hisataka.e3@f.mail.nagoya-u.ac.jp (H.M.); mori.kouki.r0@s.mail.nagoya-u.ac.jp (K.M.)
3    Graduate School of Engineering, Nagoya Institute of Technology, Nagoya 466-8555, Japan; t.takahashi.966@stn.nitech.ac.jp (T.T.); k.nishiyori.691@stn.nitech.ac.jp (K.N.); morita.yoshifumi@nitech.ac.jp (Y.M.); ishibasi@nitech.ac.jp (Y.I.)
4    Institutes of Innovation for Future Society, Nagoya University, Nagoya 464-8603, Japan; t.fukuda@ieee.org
*    Correspondence: eikoath@ncgg.go.jp; Tel.: +81-562-46-2311

**Abstract:** Digital technologies ranging from biosensors to virtual reality have revolutionized the healthcare landscape by offering innovations that hold great promise in addressing the challenges posed by rapidly aging populations. To optimize healthcare experiences for older people, it is crucial to understand their user experience (UX) with digital health technologies. This systematic review, covering articles published from 2013 to 2023, aimed to explore frequently used questionnaires for assessing digital healthcare UX among older people. The inclusion criteria were original studies assessing UX in digital health for individuals aged ≥65 years. Of 184 articles identified, 17 were selected after rigorous screening. The questionnaires used included the System Usability Scale (SUS), the User Experience Questionnaire (UEQ), and the Post-Study System Usability Questionnaire. Customized questionnaires based on models such as the Technology Acceptance Model and the Almere model were developed in some studies. Owing to its simplicity and effectiveness in assessing digital health UX among older people, the SUS emerged as the go-to tool (52.9%). Combining the SUS with the UEQ provided comprehensive insights into UX. Specialized questionnaires were also used, but further research is needed to validate and adapt these tools for diverse cultural contexts and evolving technologies.

**Keywords:** digital health technology; older people; user experience; questionnaire; system usability scale





## 1. Introduction

The healthcare landscape has undergone a profound transformation with the advent of digital technologies. These advancements encompass a wide spectrum of innovations, including biosensors, wearables [1,2], digital healthcare applications such as health apps [3–6] and chatbots [7,8], remote clinical management tools [9–13], integrated machine learning algorithms aiding decision-making [14,15], and immersive experiences such as virtual reality [16–19] and augmented reality [20–22]. Moreover, electronic medical records [23,24] and visual analytics/dashboards [25–28] have become integral components of modern healthcare, forming a digital ecosystem that presents unprecedented opportunities to enhance patient care and monitoring [22]. As far as we know, there is no general system for digital health technology, but it includes the components outlined in Table 1 from our previous research. Amidst the challenges posed by global aging, these digital health solutions hold promise in addressing issues associated with reduced physical and cognitive function, multiple chronic conditions, and shifts in social networks [29]. The

application of digital technologies has the potential to not only improve quality of life and well-being, but also foster independent living among older people.

**Table 1.** General system components of digital healthcare technologies.

| Components | Description |
| --- | --- |
| User Interface | This is the front-end component accessible through web or mobile applications that allows users to interact with the digital health system. |
| Data Collection | This component includes various sensors and devices that collect health-related data, such as wearable fitness trackers, medical devices, and IoT sensors. |
| User Profile Management | It manages user accounts, preferences, and personal information securely. |
| Authentication and Authorization | Ensures secure access to the system by verifying user identities and granting appropriate permissions. |
| Health Data Storage | Stores the collected health data securely, often in compliance with healthcare data privacy regulations. |
| Data Analysis and Processing | This component processes and analyzes health data to derive insights, identify patterns, and provide personalized recommendations. |
| Telemedicine and Communication | Facilitates real-time communication between healthcare providers and users through video calls, chat, or other telehealth services. |
| Machine Learning Algorithms | Utilizes machine learning and AI algorithms for tasks like disease diagnosis, rick prediction, and treatment recommendation. |
| Electronic Health record (EHR) System | Stores and manages users' electronic health records, including medical history, diagnoses, medications, and treatments. |
| Alerts and Notifications | Sends alerts, reminders, and notifications to users for medication, appointments, or other health-related activities. |

In light of advancements in digital health technologies and demographic shifts, it has become imperative to comprehensively investigate the user experience (UX) of these technological innovations. UX is defined as a "person's perception and responses resulting from the use and/or anticipated use of a product, system, or service" [30]. UX includes all of the user's emotions, beliefs, preferences, perceptions, physical and psychological responses, behaviors, and accomplishments that occur before, during, and after use. Usability, when interpreted from the perspective of the user's personal goals, can include the kind of perceptual and emotional aspects typically associated with UX [30]. The UX investigation of digital health technologies is a crucial step toward understanding specific needs, such as those of older people, and achieving optimal healthcare experiences.

Various methods have been used to investigate the UX of digital health technologies, encompassing questionnaires, interviews, and observations [31]. The focus of this systematic review is on questionnaires as a method for investigating the UX of digital health technologies among older people. Questionnaires provide structured and quantifiable data which can help gather a comprehensive overview of the experiences of older people with digital health technologies. Moreover, validity and reliability are crucial factors in ensuring the scientific rigor of such investigations. Original questionnaires are vital when evaluating specific aspects of digital health technologies [32]. Consequently, both standardized and custom-designed questionnaires tailored to the nuances of digital health technologies play a crucial role in uncovering the UX of older people.

Given this background, the present study aimed to review and introduce the questionnaires frequently used to assess the UX of older people when using digital healthcare technologies and to explore the contents of original or customized questionnaires. The outcomes of this study are expected to provide guidance for designing and implementing future research endeavors. Moreover, these findings are anticipated to be useful for people in the industry to assess the UX of older people. This, in turn, will significantly contribute to the development of user-centric healthcare technologies.

## 2. Materials and Methods

The methodology employed in this systematic review adhered to the Preferred Reporting Items for Systematic Reviews and Meta-Analyses (PRISMA) guidelines [33]. This systematic review has been registered in the PROSPERO International Prospective Register of Systematic Reviews (registration number: CRD42023418271). This systematic review of the literature was conducted in April 2023, utilizing data from PubMed, Google Scholar, and the Ichushi Web of Japanese journal databases. To ensure the inclusion of the latest evidence in the field, the present review analyzed manuscripts and articles published from 1 January 2013 to 31 March 2023. The selection of this time period was based on the progress and innovation observed in the field of digital health technology, as well as the corresponding brush-up on UX research methods. In formulating the inclusion criteria, the PICOS (P = population, I = interventions, C = comparator, O = outcome, S = study design) format was adopted.

The inclusion criteria were as follows: (1) original research papers that utilized questionnaires to assess the UX of digital health technologies among older people; (2) papers describing the content of original or customized questionnaires used for UX evaluation; (3) papers with main targets aged $\geq 65$ years. The exclusion criteria were as follows: (1) treview articles, protocols, conference papers, and reports; (2) papers lacking full-text availability; (3) papers that employed interviews for UX assessment; and (4) papers that assessed the UX of technologies related to specific diseases, such as cancer, surgical and radiological technologies, and oral hygiene. The targeted population for this review had no specific income-based restrictions.

The search strategy employed four categories of keywords (Table 2), with the keywords within each category combined using the "OR" Boolean operator. Subsequently, to retrieve relevant papers, the results of these searches were combined using the "AND" Boolean operator. The search was limited to the Title/Abstract search field and filtered for articles available only in English or Japanese.

**Table 2.** Keywords used for the search strategy.

| Keywords | | | |
| --- | --- | --- | --- |
| Digital health technology Digital health device | User needs User experience Quality of experience | Evaluation Assessment | Questionnaire Scale |

Following the initial search, a total of 174 articles were identified from PubMed, with an additional 6 from Google Scholar and 4 from Ichushi Web. These findings underwent analysis and screening conducted by three experts from the research team, consisting of a rehabilitation medicine researcher and two engineering researchers. This collaborative effort allowed us to consider both geriatric and technical perspectives. The initial screening was based on the titles and abstracts of the identified articles. Subsequently, the same experts evaluated the full text of selected papers. In the case of conflicting opinions, consensus was reached through discussion among the three experts. Finally, a list of included papers was compiled.

Data extraction involved collecting the following information from included papers: the first author's name, year of publication, evaluation questionnaire, details of the original

or customized questionnaire employed, and the digital health technology under assessment. The collected data were analyzed using descriptive statistics, including frequency and frequency percentages. Additionally, an investigation was conducted to determine whether the reliability and validity of the extracted questionnaires had been verified, whether manuals for these questionnaires were available, and whether benchmarks had been established.

## 3. Results

The comprehensive search of the PubMed, Google Scholar, and Ichushi Web databases initially yielded 184 papers. A thorough screening of the titles and abstracts led to the exclusion of 141 papers. Subsequently, the full texts of the remaining 43 papers underwent meticulous review. In the final analysis, 17 papers were considered suitable for data extraction [34–50] (Figure 1).

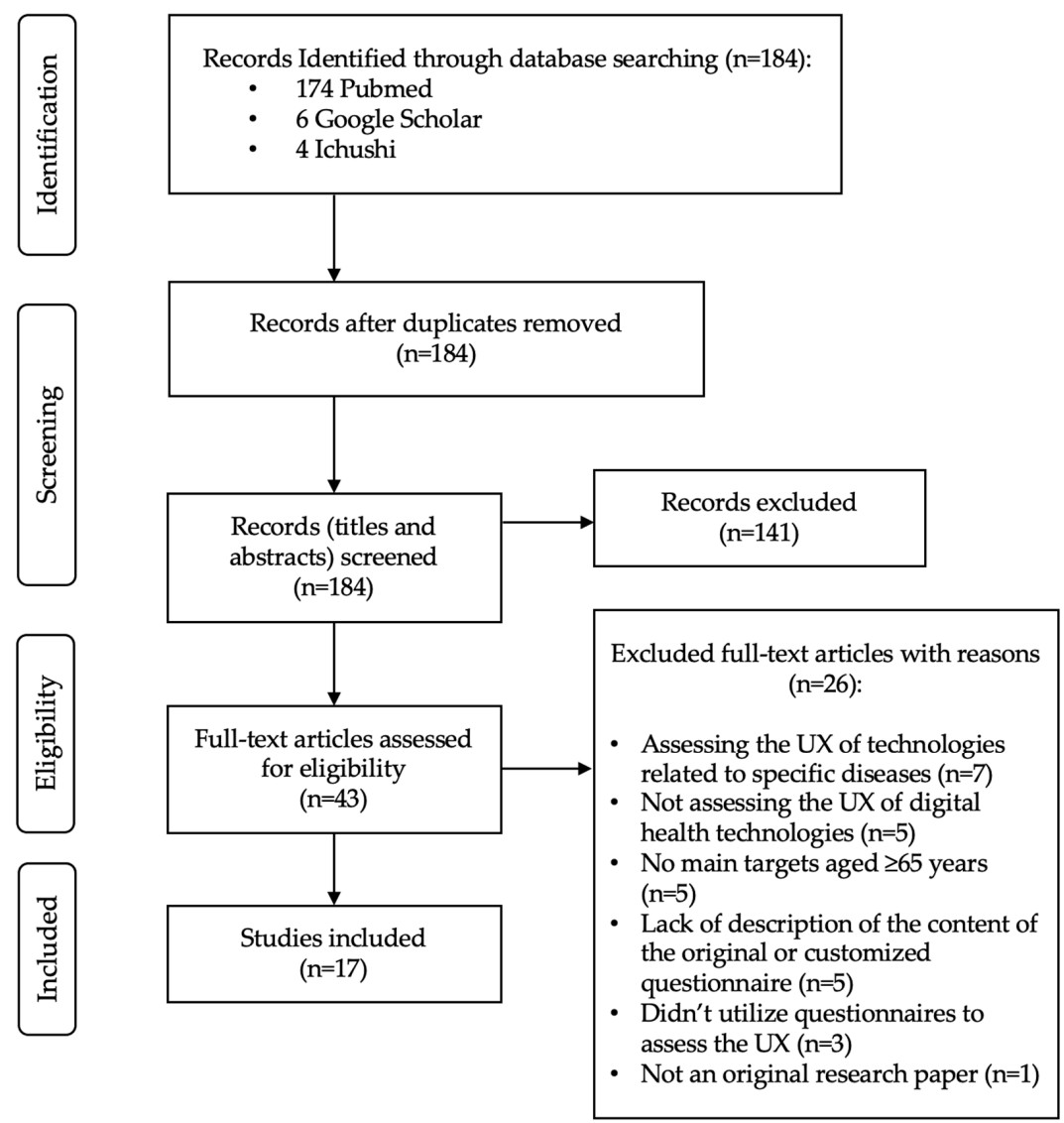

**Figure 1.** Flowchart of search strategy.

The System Usability Scale (SUS) emerged as the most frequently used standardized questionnaire for assessing digital health technologies among older people (*n* = 9, 52.9%) [35,38,39,42,44,45,48–50]. Other commonly utilized questionnaires included the User Experience Questionnaire (UEQ) and its short version (*n* = 5, 29.4%) [35,37,42,44,45],

the Post-Study System Usability Questionnaire (PSSUQ; *n* = 2, 11.8%) [34,42], the Quebec User Evaluation of Satisfaction with Assistive Technology (QUEST; *n* = 2, 11.8%) [46,48], the Facilitators and Barriers Survey/Mobility (FABS/M; *n* = 1, 5.9%) [46], the Psychosocial Impact of Assistive Devices Scale (PIADS; *n* = 1, 5.9%) [46], the Telehealth Satisfaction Questionnaire for Wearable Technology (TSQ-WT; *n* = 1, 5.9%) [47], and the User Satisfaction Evaluation Questionnaire (USEQ; *n* = 1, 5.9%) [38]. Eight studies utilized multiple standardized questionnaires [35,38,39,42,44–46,48] (Table 3).

**Table 3.** Descriptive analyses of the 17 papers included in the present review.

| Author | Year | Location | Subjects | Questionnaires | Digital Health Technology |
|---|---|---|---|---|---|
| Bakogiannis et al. [34] | 2021 | Greece | 14 patients with heart failure (mean age, 64.9 ± 9.7 years) | PSSUQ | mHealth app (the Hellenic Educational Self-case and Support Heart Failure app [ThessHF app]) |
| Bergquist et al. [35] | 2020 | Norway | 20 community-dwelling adults (mean age, 68.7 ± 5.2 years) | SUS UEQ | 3 smartphone app-based self-tests of physical function |
| Borda et al. [36] | 2018 | Australia | 96 older adults living independently (aged ≥ 55 years) | Customized questionnaire based on the TFI | Wearable device that collect health data such as heart rate, respiration rate, blood pressure, activity (steps, calories), sleep, mood, and diet |
| Chen et al. [37] | 2020 | China | 25 elders (mean age, 71.5 ± 4.1 years) | UEQ-S | AR-based exergame system to reduce fall risk |
| Domingos et al. [38] | 2022 | Portugal | 110 community-dwelling older adults (mean age, 68.4 ± 3.1 years) | Customized questionnaire based on the TAM SUS USEQ | Wearable activity tracker (Xiaomi Mi Band 2) |
| Doyle et al. [39] | 2021 | Ireland and Belgium | 120 older persons with multimorbidity (mean age, 74.2 ± 6.4 years) | SUS | Digital platform to Support Self-management of Multiple Chronic Conditions (ProACT) |
| Huang et al. [40] | 2021 | Taiwan | 29 older people living independently | Customized questionnaire based on the TAM and the Almere models | Buddy robot (emotional companion-type robot) |
| Lee et al. [41] | 2017 | South Korea | 313 adults aged >40 years | Original questionnaire based on the relevant literature | mHealth application |
| Macis et al. [42] | 2020 | Italy | (1) 40 patients aged >65 years (2) 19 older adults (mean age, 73 ± 6 years) | (1) SUS, PSSUQ (2) SUS, UEQ | Tele-social-care platform |
| Moyle et al. [43] | 2022 | Australia | 133 older adults aged ≥65 years | Customized questionnaire based on the TAM SUS UEQ | Aged care technology |
| Pérez-Rodríguez et al. [44] | 2020 | Spain | 42 inpatients aged ≥45 years | Customized questionnaire based on the TAM SUS UEQ | FriWalk robotic walker |
| Pérez-Rodríguez et al. [45] | 2021 | Spain | 37 older adults (mean age, 82.1 ± 5.4 years) | Customized questionnaire based on the TAM | Technological ecosystem for remote follow-up |
| Salatino et al. [46] | 2016 | Italy | 79 participants (including 25 aged 60–80 and 4 aged >80 years) | QUEST 2.0 PIADS FABS/M | Powered wheelchair |
| Schmidt et al. [47] | 2022 | Germany | 80 older adults (mean age, 67 ± 4 years) | TSQ-WT | Commercially available activity trackers (Fitbit) |

**Table 3.** *Cont.*

| Author | Year | Location | Subjects | Questionnaires | Digital Health Technology |
|---|---|---|---|---|---|
| Stara et al. [48] | 2018 | Ireland Italy Israel | 15 older adults (mean age, 70.0 ± 6.4 years) | QUEST SUS | WIISEL (Wirelenn Insole for Independent and Safe Elderly Living) system that monitor fall risk and to detect falls |
| Sun et al. [49] | 2019 | USA | 29 female participants (mean age, 77.5 ± 7.9 years) | SUS | Kinect camera-based self-initiated fall risk assessment tool |
| van Velsen et al. [50] | 2018 | Netherlands | 24 older people (mean age, 71.6 years; range, 62–87 years) | SUS | Tablet technology for screening for health |

**AR**, augmented reality; **FABS/M**, Facilitators and Barriers Survey/Mobility; **PIADS**, Psychosocial Impact of Assistive Devices Scale; **PSSUQ**, Post-Study System Usability Questionnaire; **QUEST**, Quebec User Evaluation of Satisfaction with Assistive Technology; **SUS**, System Usability Scale; **TAM**, Technology Acceptance Model; **TFI**, Tilburg Frailty Indicator; **TSQ-WT**, Telehealth Satisfaction Questionnaire for Wearable Technology; **UEQ**, User Experience Questionnaire; **USEQ**, User Satisfaction Evaluation Questionnaire.

Table 4 describes the standardized questionnaires used to assess the UX of digital health technologies among older people and provides information on their reliability and validity, the presence of manuals, and benchmarks.

Among the seventeen studies identified, seven utilized original or customized questionnaires [36,38,40,41,43–45], of which four relied exclusively on customized questionnaires. [6,10,11,13]. Of the six original or customized questionnaires, four were developed based on the Technology Acceptance Model (TMT) [38,43–45], one was created incorporating elements of both the TMT and the Almere model [40], another was derived from the Tilburg Frailty Indicator (TFI) [36], and one was created using relevant literature related to healthcare services, mobile services, and information communication services [41] (Table 3).

**Table 4.** The standardized questionnaires used to assess the user experience of digital health technologies among older people.

| Questionnaire | Frequency *n*, (%) | Description of Questionnaire | Construct Validity | Criterion-Related Validity | Other Validities | Reliability | Manual | Benchmark |
|---|---|---|---|---|---|---|---|---|
| SUS [51] | 9 (52.9) | Developed by John Brooke in 1986. It assesses the usability of various products and services such as hardware, software, mobile devices, websites, and applications. It consists of a 10-item questionnaire on a 5-point Likert scale from "Strongly agree" to "Strongly disagree". The scores range from 0 to 100, with scores >68 indicating good usability [51,52]. | Strong correlations among the selected items were reported, with absolute values of r ranging from 0.7 to 0.9 [51]. | Not verified | Bangor et al. [53] found significant concurrent validity with a single 7-point rating of user-friendliness (r = 0.806). | The reliability of SUS was at or just over 0.90 [53]. | (Calculator and guide) https://measuringu.com/product/suspack/, accessed on 10 July 2023. | A score > 68 would be considered above average and anything < 68 is below average. Grading scale: F, SUS score < 60; D, 60 ≤ score < 70; C: 70 ≤ score < 80; B: 80 ≤ score < 90; A: score ≥ 90 [52]. |
| UEQ [54] | 5 (29.4) | The UEQ uses 26 pairs of contrasting adjectives to evaluate six aspects—attractiveness, perspicuity, efficiency, dependability, stimulation, and novelty—with scales from –3 (most negative answer) to +3 (most positive answer) [54]. It includes a comprehensive analysis tool and benchmark data set, enabling insightful product quality assessment based on a wide range of user experience studies [55]. | According to a factor analysis, the responses to the questionnaire did not align with the initially designed structure [56]. | The high correlation between the UEQ dimensions and the SUS (r between 0.60 and 0.82, *p* < 0.0001) confirmed the validity of this questionnaire [57]. | Not verified | The reliability of the UEQ was confirmed with a high Cronbach's alpha score of 0.98 [57]. | (Handbook) https://www.ueq-online.org/Material/Handbook.pdf, accessed on 10 July 2023. | General Benchmark Business Software Websites and Web services https://www.ueq-online.org, accessed on 10 July 2023. |
| PSSUQ Version 3 [58] | 2 (11.8) | PSSUQ Version 3, a 16-item tool on a 7-point Likert scale between "Strongly agree" and "Strongly disagree", assesses user satisfaction with a website, software, or product post-study. Lower scores indicate better performance and satisfaction. It has three subscales: System Quality (SysQual), Information Quality (InfoQual), and Interface Quality (IntQual), each derived from the average scores of respective question sets [59]. | The factor analysis revealed that three factors accounted for 87% of the total variance [58]. | It showed a moderate correlation (r = 0.80) with other measures of user satisfaction [58]. | It has shown evidence of concurrent validity [58]. | The earliest versions of the PSSUQ demonstrated significant reliability across both the overall scale and its subscale. For Version 3, the reliability coefficients are as follows: Overall: 0.94, SysQual: 0.90, InfoQual: 0.91, IntQual: 0.83 [59]. | Not applicable | SYSUSE: 3.80 INFOQUAL: 3.02 INTERQUAL: 2.49 Overall: 2.28 https://uiuxtrend.com/pssuq-post-study-system-usability-questionnaire/, accessed on 10 July 2023. |

**Table 4.** *Cont.*

| Questionnaire | Frequency n, (%) | Description of Questionnaire | Construct Validity | Criterion-Related Validity | Other Validities | Reliability | Manual | Benchmark |
|---|---|---|---|---|---|---|---|---|
| QUEST 2.0 [60] | 2 (11.8) | It consists of 12 items evaluating user satisfaction with the product (8 items) and service delivery (4 items). Each item is posed as a satisfaction query (e.g., "How satisfied are you with the <questionnaire item> of your assistive device?" (e.g., "How satisfied are you with the ease of adjusting the parts of your assistive device?") and uses a 5-point Likert scale from "not satisfied at all" to "very satisfied" [60]. | Not verified | Positive correlations were found between QUEST 2.0 and the three Psychosocial Impact of Assistive Devices Scale (PIADS) dimensions. They were fair to moderate for the Device and total QUEST (Pearson correlation coefficient (r between 0.34 to 0.45) and fair with Services (r between 0.27 to 0.30) [60]. | Not verified | The test-retest reliability, as assessed using the intraclass correlation coefficient, yielded values of 0.82, 0.82, and 0.91 [60]. | Not applicable | Not applicable |
| FABS/M [61] | 1 (5.9) | It examines how environmental factors affect the daily use of assistive devices, acting as facilitators or barriers. It is composed of 133 questions grouped into six domains: primary mobility device, home-built features, community-built environment and natural features, community destination access, community facilities access, and community support network [61]. | Assessment of construct validity was conducted [62]. | Not verified | Assessment of internal consistency, content validity, and face validity was conducted [62]. | The items encompassed within the community support network domains exhibit moderate infernal consistency, as reflected by a Cronbach's alpha value ranging from 0.35 to 0.90. Test-retest reliability was demonstrated with Person's r values ranging from 0.52 to 0.82 [61]. | Not applicable | Normative data for SCI and Stroke [61]. https://www.sralab.org/rehabilitation-measures/facilitators-and-barriers-survey-environmental-influences-participation-among-people-lower-limb, accessed on 10 July 2023. |

**Table 4.** *Cont.*

| Questionnaire | Frequency *n*, (%) | Description of Questionnaire | Construct Validity | Criterion-Related Validity | Other Validities | Reliability | Manual | Benchmark |
|---|---|---|---|---|---|---|---|---|
| PIADS [63] | 1 (5.9) | It comprises 26 items, exploring perceived ability (12 items), adaptability (6 items), and self-esteem (8 items). Each item is presented in the format "How has the device influenced your <item>" (e.g., "How has the device influenced your openness to new experiences?") and is rated on a 7-point scale (−3 to +3), ranging from "decreased a lot/a lot worse" to "increased a lot/a lot better" [63]. | A positive correlation was found between the total score on PIADS and Pleasure and Dominance on Mehrabian and Russell's PIADS, although the specific coefficient was not reported [64]. | Not verified | Not verified | The test–retest reliability over a 2-week interval was deemed adequate, with an intraclass correlation coefficient of 0.45 [65]. | https://www.sralab.org/rehabilitation-measures/psychosocial-impact-assistive-devices#osteoarthritis, accessed on 10 July 2023. | Not exist |
| TSQ-WT [66] | 1 (5.9) | It assesses user satisfaction with wearable technologies. It encompasses six dimensions that evaluate the benefits, usability, self-concept, privacy, loss of control, quality of life, and wearing comfort of a system. Each dimension contains five items rated on a 5-point Likert scale from 0 ("not at all") to 4 ("fully agree"), with higher scores indicating more positive ratings [66]. | Unpublished | Unpublished | Unpublished | Unpublished | Unpublished | Unpublished |
| USEQ [67] | 1 (5.9) | It assesses user satisfaction, a component of usability, in virtual rehabilitation systems. Comprising six items rated on a 5-point Likert scale, scores range from 6 ("poor satisfaction") to 30 ("excellent satisfaction"). The score is evaluated using the following classifications: poor (0–5), fair (5–10), good (10–15), very good (15–20), or excellent (20–25) [67]. | Not verified | Not verified | Not verified | The six items of the USEQ were significantly correlated, demonstrating good internal consistency with a Cronbach's alpha coefficient of 0.716 [67]. | Not applicable | Not applicable |

**FABS/M**, Facilitators and Barriers Survey/Mobility; **PIADS**, Psychosocial Impact of Assistive Devices Scale; **PSSUQ**, Post-Study System Usability Questionnaire; **QUEST**, Quebec User Evaluation of Satisfaction with Assistive Technology; **SUS**, System Usability Scale; **TSQ-WT**, Telehealth Satisfaction Questionnaire for Wearable Technology; **UEQ**, User Experience Questionnaire; **USEQ**, User Satisfaction Evaluation Questionnaire.

## 4. Discussion

According to the findings of this review, the SUS emerged as the most used standardized questionnaire to assess the UX of digital health technologies among older people, and it was used without modifications in its standard form. Despite being a self-described "quick and dirty" usability scale, the SUS has garnered widespread popularity in usability assessments [51,68,69]. It comprises 10 items, each offering five scale steps, and includes relevant questions that produce findings useful in understanding UX, with odd-numbered items conveying a positive tone and even-numbered items expressing a negative tone. Participants are instructed to complete the SUS immediately after interacting with the digital health technologies, recording their prompt responses to each item. The SUS generates an overall score range from 0 to 100 in 2.5-point increments, and online tools for calculating SUS scores are freely available. No license fee is required for using the SUS; the only prerequisite is acknowledgment of the source. Various translations of the SUS exist, but only a few have undergone psychometric evaluation [70]. Recent studies have consistently demonstrated high reliability for the SUS (approximately 0.90) and established its validity and sensitivity in discerning usability disparities, interface types, task accomplishment, UX, and business success indicators [53]. Furthermore, its concise 10-item format makes it user-friendly, especially for older people. Additionally, the global average score for overall SUS is $68.0 \pm 12.5$ [53]. The grading scale is as follows: F, SUS score < 60; D, $60 \leq$ score < 70; C, $70 \leq$ score < 80; B, $80 \leq$ score < 90; A, score $\geq$ 90 [52]. This grading scale is structured in a way that considers a SUS score of 68 as the center of the range for the 'C', such as grading on a curve. Furthermore, the high correlation between the SUS and the UEQ (r between 0.60 and 0.82, $p < 0.0001$) [57] could further emphasize its potential as a standard tool for evaluating usability and UX in digital healthcare technologies for older people.

The UEQ was the second most frequently employed standardized questionnaire. Furthermore, the UEQ was utilized in conjunction with the SUS in four of the five studies it was employed in [35,42,44,45]. The UEQ and SUS serve similar purposes in assessing the UX and usability of digital products, but have some differences and unique advantages. The UEQ includes six aspects—attractiveness, perspicuity, efficiency, dependability, stimulation, and novelty [54]—and offers a more comprehensive and detailed assessment of UX, including emotional and experiential aspects. On the other hand, the SUS is a simple and effective tool for assessing overall usability [51,68,69]. It focuses on the usability aspect of UX, providing a single score that indicates overall usability [51]. Bergquist et al. [35] assessed the usability of three smartphone app-based self-tests of physical function in home-based settings using the SUS and UEQ, and reported that the participants experienced high levels of perceived ease of use when using the apps. Macis et al. [45] employed the SUS and UEQ to assess the usability and UX of a novel mobile system to prevent disability; in particular, the UEQ was included as an evaluation tool to complement the domains addressed by the SUS. Thus, combining both the UEQ and SUS might provide a more holistic understanding of UX.

The following additional questionnaires were also used: PSSUQ, QUEST, FABS/M, PIADS, TSQ-WT, and USEQ. Only TSQ-WT was used individually. The PSSUQ focuses on assessing system usability and measures factors such as system performance, reliability, and overall satisfaction [58,59]. The QUEST evaluates user satisfaction with assistive technology specifically and considers factors such as the device's effectiveness, ease of use, and impact on daily life [60]. The FABS/M is designed to assess mobility-related assistive devices and examines the facilitators and barriers that users encounter when using these devices [61]. The PIADS measures the psychosocial impact of assistive devices, focusing on how these devices affect users' self-esteem, well-being, and participation in daily activities [63]. The TSQ-WT is tailored for assessing user satisfaction with wearable telehealth technologies, considering factors such as ease of use and satisfaction with remote health monitoring [66]. The USEQ is a general questionnaire for evaluating user satisfaction without focusing on specific domains [67]. These questionnaires are specialized for different purposes, such as system usability, assistive technology, mobility devices, psychosocial impact, telehealth,

and overall user satisfaction. Researchers choose the more relevant ones based on study objectives and the technology being assessed.

The findings of this review indicate that six customized questionnaires have been developed based on the TMT, Almere model, and TFI. The TAM, initially formulated by Davis in 1989 [71], is a widely recognized theoretical framework in the field of technology adoption and user acceptance. It aims to clarify the factors that influence an individual's decision to accept and use a new technology or information system. Its core components are perceived usefulness and perceived ease of use, and these factors directly influence a user's intention to use technology, which in turn affects their actual usage behavior [71]. It has been used as a basis for developing questionnaires to assess user perceptions of new technologies [38,40,43–45]. These questionnaires, which aim to measure perceived usefulness and perceived ease of use, help predict whether users will adopt a technology.

On the other hand, the Almere model functions as a conceptual framework in the field of UX design and research. It is designed to provide a structured approach to understanding and enhancing the UX of digital products and services, especially concerning users' acceptance of socially assistive robots [72]. It encompasses 12 constructs, as stated in the introduction part [72]. The present review reveals that the Almere model has been used to measure attitudes toward robot assistance in regard to activities of daily living among older people living independently [40].

The TFI is an instrument used to quantify the frailty of older adults from their own perspective across three domains: physical, psychological, and social [73]. Borda et al. [36] employed the TFI to gain insights into health self-management requirements, frailty, age-related changes, and the health information support provided by consumer wearable devices, particularly among older adults living independently.

Additionally, Lee et al. [41] adapted the survey questionnaire items from relevant literature on healthcare, mobile, and information communication services to explain the intentions of consumers regarding the use of mHealth applications. This comprehensive assessment encompassed seven key constructs, each measured with multiple items. The scale items for health stress and the epistemic value as contextual personal values were adapted from Sheth et al. [74], while the measurement items for emotion (enjoyment) and intention to adopt were developed based on the work of Davis et al. [71]. In addition, the items measuring reassurance were taken from O'Keefe and Sulanowski [75] and modified for the study. The convenience value was borrowed from Berry et al. [76] and developed in the study. Finally, the measurement items for the usefulness value were developed with reference to Davis's items [71]. All of the measurement items were modified to emphasize health-related events and behaviors based on personal values with respect to mHealth [77,78]. They also included the contents of mHealth services. Each item was measured on a five-point Likert scale ranging from "Strongly disagree" to "Strongly agree".

Customized questionnaires have been developed based on the theoretical frameworks of the TMT, Almere model, and TFI. These frameworks primarily focus on understanding the acceptability of digital technology rather than the user experience. This distinction is crucial because acceptability assessments primarily investigate factors influencing the decision to adopt and use technology, such as perceived usefulness and ease of use, which are essential for technology adoption. On the other hand, user experience assessments delve deeper into the subjective aspects of interaction with technology, including emotional responses, satisfaction, and usability. Seknoh et al. (2022) reported that they developed a generic theoretical framework of acceptability questionnaires that can be adapted to assess the acceptability of any healthcare intervention [79]. Assessing acceptability might help identify characteristics of interventions that may be improved.

In conclusion, this systematic review highlights the prevalent use of standardized questionnaires for evaluating the UX of digital health technologies among older people. In these questionnaires, the SUS emerged as the most frequently employed tool, underscoring its popularity and effectiveness in assessing usability. Its simplicity, wide applicability, and availability of benchmarks make it a valuable choice for evaluating digital healthcare

technologies. Additionally, the high correlation observed between the SUS and UEQ suggests their complementary use for a comprehensive understanding of UX. Furthermore, several other specialized questionnaires were utilized, each tailored to assess specific aspects of usability and user satisfaction, including user acceptability as a main focus based on theoretical frameworks. These tailored approaches emphasize the importance of aligning UX assessment with theoretical frameworks and considering user acceptability, providing a holistic approach to evaluating digital health technologies for older people.

This study has several limitations. First, the scope of this review was limited to studies published in English or Japanese, potentially excluding valuable research conducted in other languages. Moreover, the exclusion of papers lacking a full text and those employing interviews for UX assessment might have led to the omission of relevant studies. Second, the focus on older people might limit the generalizability of the findings to younger populations. Third, the evolving landscape of digital health technologies necessitates continued research and updates to encompass the latest developments. Finally, the psychometric properties of some questionnaires and their applicability to diverse cultural contexts require further investigation to enhance their validity and reliability in assessing UX among older people.

**Author Contributions:** E.T. designed this study, conducted the data collection, and drafted the manuscript. H.M. performed the data analysis and provided critical revisions to the manuscript. T.T., K.N. and K.M. assisted in the data collection, conducted the literature review, and contributed to the manuscript editing. Y.M., T.F. and I.K. provided expertise in digital health technologies and reviewed and edited the manuscript for subject-specific content. Y.I. reviewed and critiqued the manuscript, provided valuable intellectual input, and supervised the project. All authors have read and agreed to the published version of the manuscript.

**Funding:** This research is based on data collected for a project funded by the "Knowledge Hub Aichi", Priority Research Project IV from the Aichi Prefectural Government in September 2022.

**Data Availability Statement:** No new data were created or analyzed in this study. Data sharing is not applicable to this article.

**Conflicts of Interest:** The authors declare no conflict of interest.

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
