# Peer review of "User Experience of Older People While Using Digital Health Technologies: A Systematic Review"

_applsci, doi:10.3390/app132312815_

Round 1

Reviewer 1 Report

Comments and Suggestions for Authors

The area of survey is interesting. However, this paper can be improved by considering following points.

1. How about including general system diagram for the digital health technology showing its different components?

2. Please define user-centric healthcare technology and discuss on some of the representative approaches to address it. 

3. In the discussion section, please present a comparative table for different tools you analyzed. Compare them on the basis of the domain specific relevant parameters, and rank them. It makes the idea you presented more clear.

4. Since usability and satisfaction are subjective perceptions they can't be generalized. How would you address them?

5. If possible, please show that how this review is different from other review process. Any new element/s have been added to add the value of the work?

Comments on the Quality of English Language

Minor proof reading is necessary.

Author Response

Thanks to the insightful comments from the reviewers, we were able to make significant improvements to the paper. The revisions have been highlighted in blue text in the manuscript. Additionally, we have documented them in a PDF in a question-and-answer format. We appreciate the valuable feedback from the reviewers.

Reviewer 2 Report

Comments and Suggestions for Authors

Dear authors, 

Thank you for submitting your work on this topic. I enjoyed reading it and I see the value of it. I have attached additional comments to help revise your paper. 

Thanks 

Author Response

(The authors gave the same response as above.)
